# Motivation, Usability, and Credibility of an Intelligent Activity-Based Client-Centred Training System to Improve Functional Performance in Neurological Rehabilitation: An Exploratory Cohort Study

**DOI:** 10.3390/ijerph18147641

**Published:** 2021-07-18

**Authors:** Els Knippenberg, Ilse Lamers, Annick Timmermans, Annemie Spooren

**Affiliations:** 1Department of Healthcare, Centre of Expertise–Innovation in Care, PXL University of Applied Sciences and Arts, 3500 Hasselt, Limburg, Belgium; Annemie.Spooren@PXL.be; 2Faculty of Rehabilitation Sciences, REVAL, Hasselt University, 3590 Diepenbeek, Limburg, Belgium; Annick.Timmermans@uhasselt.be (A.T.); Ilse.Lamers@Noorderhart.be (I.L.); 3Noorderhart, Rehabilitation and MS Center, 3900 Pelt, Limburg, Belgium

**Keywords:** client-centred, task-oriented, neurorehabilitation, technology, motivation

## Abstract

(1) Background: technology-based training systems are increasingly integrated in neurorehabilitation but are rarely combined with a client-centred task-oriented approach. To provide a low-cost client-centred task-oriented system, the intelligent activity-based client-centred task-oriented training (i-ACT) was developed. The objective was to evaluate the usability, credibility and treatment expectancy of i-ACT, and the motivation towards i-ACT use in rehabilitation over time. Additionally, this study will evaluate the upper limb treatment effects after training with i-ACT. (2) Methods: a mixed-method study was performed in four rehabilitation centres. Training with i-ACT was provided during six weeks, three times per week, forty-five minutes per day, additional to conventional care. (3) Results: seventeen persons with central nervous system diseases were included. High scores were seen in the system usability scale (score ≥ 73.8/100), credibility (score ≥ 22.0/27.0)/expectancy (score ≥ 15.8/27.0) questionnaire, and intrinsic motivation inventory (score ≥ 5.2/7.0), except the subscale pressure (score ≤ 2.0/7.0). Results from the interviews corroborate these findings and showed that clients and therapists believe in the i-ACT system as an additional training support system. Upper limb functional ability improved significantly (*p* < 0.05) over time on the Wolf motor function test. (4) Conclusion: i-ACT is a client-centred task-oriented usable and motivational system which has the potential to enhance upper limb functional training in persons with neurological diseases.

## 1. Introduction

Central nervous system diseases such as stroke, multiple sclerosis, and spinal cord injury are common causes of physical, cognitive and/or sensory disability worldwide. Impairments in both upper and lower extremities and trunk control affect the ability to perform activities of daily life (ADL), and consequently impede the quality of life (QoL) of neurological patients [1,2,3]. Studies have already established the importance of rehabilitation for neurological patients to regain functional ability in upper extremities [4,5,6,7]. Furthermore, the relevance and benefits of a client-centred task-oriented approach in all stages after injury have been indicated in neurorehabilitation [8,9,10,11]. A client-centred approach implies not only to incorporate patient’s wishes and needs, but also actively involve the patient in setting goals for their rehabilitation process. Task-oriented means that their rehabilitation incorporates specific, functional tasks based on task segmentation. Furthermore, the training components such as random and distributed practice, exercise progression, feedback and clear functional goals can enhance the outcomes [8,9,10]. Nevertheless, implementation of a client-centred task-oriented approach in clinical practice is difficult, as it may be more time consuming for therapists and therefore costly for the rehabilitation centres [10,12]. Additionally, therapists have limited time per patient. Technological innovations could provide a solution to counteract these issues. Robotic systems have already shown their value in providing extra therapy time in rehabilitation [13,14,15,16,17,18], but at the moment, very few systems allow task-oriented training, and no system, at present, includes a client-centred approach [13,17,19]. Furthermore, the cost of robotic systems is mostly quite expensive, especially when applied at home [19,20]. Sensor-based systems that are easy to use and low-cost, such as the Nintendo Wii and Microsoft Kinect^®^ for Xbox 360, are promising solutions but have limited utility in rehabilitation for impaired populations [21,22]. In clinical research, patients show to be more motivated to perform rehabilitation exercises with these new technologies, and their adherence to treatment is greater [21,23,24,25,26,27]. Furthermore, the independent use of these technologies by patients might lead to increased therapy intensity. However, to date, these commercially available systems do not incorporate the client-centred and task-oriented approach, and the standard (exer)games are not developed to meet rehabilitation goals such as coordination patterns, compensation strategies, etc. when performing task-oriented exercises [20,21]. The sensors, however, show promising features for use in rehabilitation, such as skeleton tracking in Microsoft Kinect^®^. Therefore, the authors explored these features in earlier research [28] and developed an intelligent activity-based client-centred training (i-ACT) system using the Microsoft Kinect^®^ sensor and software development kit (SDK). With i-ACT, therapists can record a movement based on the individually set goals of a persons with CNS central nervous system disorder. Next, it is possible to implement certain training parameters such as the number of repetitions, amount of rest, number and location of the targets. During training, the patient has a mirrored view of themself represented as an avatar and is stimulated to perform the tasks or activities presented on the screen. A more detailed description is provided in the method section. The i-ACT has been developed in co-creation with clients and therapists and integrates the client-centred task-oriented approach. The aim of this study was (1) to gather information on the usability, credibility and treatment expectancy of i-ACT, and patients’ motivation towards i-ACT during six weeks of training and short-term follow-up, (2) to explore the effect of additional i-ACT training on functional ability and perceived performance of persons with neurological diseases, and (3) to gather users’ feedback on additional i-ACT training.

## 2. Materials and Methods

### 2.1. Participants

A homogeneous convenience sample [29] was recruited between January 2016 and August 2016 among persons with central nervous system disorders in four rehabilitation centres in Flanders (Belgium). The inclusion criteria were: age over 18 years old, a medical diagnosis of central nervous system disease, and a dysfunction in the upper limb and/or core stability. Persons with multiple sclerosis (MS) had to be free of treatment with corticosteroids for one month so that possible functional improvement is not related to the corticosteroids. Persons with stroke or spinal cord injury had to be at least three months post-injury so that possible functional improvement could be less determined by any spontaneous recovery. Exclusion criteria were: severe spasticity (when spasticity impedes independent movement), severe cognitive impairment (person is not able to understand and follow instructions), severe communicative impairment (person is not able to answer questions), severe visual impairment (person is not able to see the television screen), and persons who use an electric wheelchair as the Microsoft Kinect^®^ might have troubles recognising a human skeleton.

Furthermore, eight occupational therapists, two from each participating centre, were recruited for participating in a semi-structured interview regarding their own experiences with the i-ACT system. The selected therapists were the most involved local healthcare professionals who observed and/or worked with the i-ACT system.

### 2.2. Study Protocol 

A mixed-method cohort study was performed. Participants received 3 × 45 min of training per week during six weeks, additional to conventional care.

The Canadian occupational performance measure (COPM) was used to collect and evaluate patients’ individual goals towards rehabilitation, e.g., drinking from a cup, drying hair with a hair dryer, lifting knees for stair for stair climbing, reaching and grasping big to small objects, etc. These goals were discussed with the therapists before implementing movements and exercises for each patient in the i-ACT system. The first week of the experimental intervention, the training was customised to the patients’ individual goals. Patients were asked to provide feedback on the exercises (number of repetitions, places of targets, complexity, etc.). If needed, movements and exercises were adapted so that patients could be successful, but also challenged. When exercises were performed successfully (i.e., all targets reached without compensational movements) and/or patients indicated that the exercise was too monotonous, more challenging exercises were provided.

The following demographic data were obtained from the medical files: age, gender, diagnosis, and time since diagnosis. Outcomes measures were collected by an independent researcher (EK) at baseline or after one training with i-ACT (T0), after 2 weeks (T1), 4 weeks (T2) and 6 weeks (T3) of training and 9 weeks after the cessation of training (T4). To provide a stable image of the participant, baseline measures were performed three times over 3 weeks (T0a, T0b and T0c). The mean of all three baseline data was used as the baseline result (T0). Table 1 provides an overview of assessments and assessment times.

A semi-structured interview was performed after training (T3) by an independent researcher (EK) to gather more information about the participants’ and therapists’ perception towards i-ACT. A topic list (see Appendix A) was formulated beforehand and interviews were recorded with an audiotape. The focus of the interviews was the experiences of both patients and therapists, advantages and disadvantages, and whether they would want to use the i-ACT system in rehabilitation.

### 2.3. Apparatus

In this study, a prototype of i-ACT was used. i-ACT consists of the Microsoft Kinect^®^ sensor and the Microsoft Kinect^®^ software development kit (SDK). For the technical development of i-ACT, the cross-platform Unity3D was used. A detailed description of the user-centred development of i-ACT is explained elsewhere [30]. The most important feature of i-ACT is that client-centredness is fully involved in every step: therapists record a movement that is valuable for the neurological patients and sets up the necessary (training) parameters to progress towards an exercise, based on clinical judgement. These parameters include the number of repetitions, target placement, etc. and can be individually set to the functional level of the neurological patient. When this is set, the neurological patient is placed in front of the i-ACT. Due to the fact that the Microsoft Kinect^®^ sensor detects a human shape, the shape and movements of the neurological patient is detected. The neurological patient is asked to perform the recorded exercise (by the therapist) by following the example avatar in a virtual environment as seen in Figure 1. The neurological patient has to copy the movement (by means of mirror image) and receives, during the performance, real-time feedback on successful trajectories and stabilisation of the body areas. In this study, the training with i-ACT was supervised by a trained therapist in a one-on-one situation.

### 2.4. Outcome Measures

The primary outcome measures were the intrinsic motivation inventory (IMI) [31,32,33], the system usability scale (SUS) [34,35], the credibility/expectancy questionnaire (CEQ) [36,37], the Canadian occupational performance measure (COPM) [38,39], and Wolf motor function test (WMFT) [40,41].

The IMI [31,32,33] is a multidimensional questionnaire to assess the participants’ subjective experience related to a target activity. The IMI consists of six subscales: interest/enjoyment, perceived competence, effort, felt pressure/tension, value/usefulness and relatedness. People score on a scale from 1 (totally not true) to 7 (totally true). Some scores need to be reversed by subtracting the response score from 8. A total IMI-score is not recommended; therefore, subscale scores are used in the analyses [31,33].

The SUS [34,35] is a questionnaire to assess the usability of a product. The item scores range from 1 (totally disagree) to 5 (totally agree) and are converted into a score from 0 (negative) to 100 (positive) [34,35]. A score of 68 or higher is considered the threshold for good usability [42,43].

The CEQ [36,37] is a questionnaire with two subscales: credibility and expectancy. Some questions are scored from 1 (totally not) to 9 (totally yes); other questions are scored with percentages (0% to 100%). The percentages are converted to scores on a scale of 1 to 9. The maximum score on each subscale is 27 (i.e., three questions per subscale). A moderate score ranges between 13.50 to 20.25; scores above 20.25 are considered high [36,37].

Next to the use of determining rehabilitation goals, the COPM is used in this study to assess the performance and satisfaction on the progress of individual goals. The COPM [38,39] is a client-centred individualised instrument. By means of a semi-structured interview, clients are asked to identify their five main goals in self-care, productivity and/or leisure. These five goals are scored on execution and satisfaction with scores ranging from 0 (negative) to 10 (positive) [38,44,45,46].

The WMFT [40,41] is a test for arm–hand functioning and contains 17 items (two strength-based tasks and 15 function-based tasks), arranged in order of complexity. The strength-based tasks are measured by weight lift and grip strength, while the 15 function-based tasks are timed and scored on a scale from 0 (not able to perform task) to 5 (normal performance) [40,41]. In this research, the 15 function-based tasks are assessed and analysed.

The secondary outcome measures were the manual ability measure-36 (MAM-36) [47], modified fatigue impact scale (MFIS) [48,49], trunk impairment scale (TIS) [50,51], and active range of motion (AROM) of the shoulder.

The MAM-36 [47] is a questionnaire that consists of 36 questions that relate to the ease or difficulty level of how a person is able to perform unilateral and bilateral ADL-tasks. Scores range from 0 (impossible) to 4 (easy) [40,47].

The MFIS [48,49] is a questionnaire which provides information on how fatigue impacts the life of the person, in terms of physical, cognitive, and psychosocial functioning. Twenty-one items are scored on a five-point Likert-scale (range from 0 = never to 4 = almost always) [48,49].

The TIS [50,51] is an assessment to measure motor impairment of the trunk by evaluating three aspects: static sitting balance, dynamic sitting balance, and trunk co-ordination. Each item is scored on a two-, three- or four-point scale, ranging from a minimum of 0 to a maximum of 23 points [50,51].

The AROM refers to the possible range in motion while performing non-assisted voluntary movement of a body part. In this study, the AROM of the shoulder joint is measured with a goniometer for abduction and flexion in relation to the torso.

The WMFT, MAM-36, TIS and AROM were tested on the most affected side.

### 2.5. Ethics Statement

All study procedures were approved by the Medical Ethics Committee of UZ KU Leuven (Registration number B322201525549) and the local ethics committees. The trial is retrospectively registered at clinicaltrials.gov (NCT04692311). All participants were informed about the study by the head researcher and were included after they gave written informed consent.

### 2.6. Statistical Analysis

For quantitative data, within-group differences of all the assessments were analysed using Friedman’s ANOVA and Wilcoxon signed-rank test (two-sided). The level of significance was set at α = 0.05. No Bonferroni correction was applied as it is not compulsory in an exploratory study; furthermore, the results are regarded as a hypothesis for further investigations. All quantitative data were analysed using SPSS software (IBM SPSS Statistics version 24, Chicago, IL, USA).

The qualitative data was collected and recorded with a voice recorder by an independent researcher (EK). Analysis was performed by two independent researchers (AS and EK) based on predefined categories as the qualitative data were used to gain more in-depth information on patients’ and therapists’ experiences on usability, motivation, credibility and expectancy towards training with i-ACT.

## 3. Results

An overview of the patient characteristics is shown in Table 2. In total, 17 patients and eight occupational therapists were included. Four patients were discharged from hospital/the rehabilitation centre during the training period (T0–T3). Three of them wished not to participate further in the i-ACT training as well as follow-up assessments T3 and T4 (see Table 3). One patient was able and willing to participate in the follow-up assessment T4. All other patients completed the full protocol. This resulted in an adherence of 89.22%. The data from the four discharged participants was treated as missing data and excluded pairwise in the data analysis.

### 3.1. Quantitative Results

Table 3 provides an overview of all results, differences over time (Friedman’s ANOVA T0–T4) and within-group differences (Wilcoxon signed-rank test) between baseline versus cessation of training (T0–T3), baseline versus follow-up (T0–T4), and cessation of training versus follow up (T3–T4).

As shown in Table 3, all subitems on the IMI scored good to very good and remained stable during the training period (T0–T3). The subscale “perceived performance” even showed an overall significant improvement over time (*p* = 0.025). Further analysis revealed that the significant improvement mainly occurs during the training period (*p* = 0.017) as was expected. The subitem “felt pressure/tension” was low, indicating that using the i-ACT was not too stressful for the patients.

Regarding usability, scores on the SUS are considered good to very good, but the scores decline slightly during training (T0 median = 77.50 (71.25–85.0) versus T3 median = 75.00 (63.75–90.00)) and short-term follow-up (median = 73.75 (64.38–92.50), but overall, they remain high (median > 68 is considered good usability [43].

Concerning the credibility of the i-ACT system for upper limb rehabilitation, patients’ scores are good to very good, and increase slightly during training (T0 median = 22.00 (18.00–25.00) versus T3 median = 23.00 (16.00–24.50)), and after a follow-up period (T4 median = 24.00 (20.25–25.00)). The patients’ expectancy towards recovery after i-ACT is moderate and increases during the training period (T0 median = 16.90 versus T3 median = 17.70 (11.25–21.00)) but declines slightly after follow-up of 9 weeks (T4 median = 15.85 (10.30–19.83)).

With respect to the arm and hand functional ability and performance time, measured by WMFT, patients showed significant improvement over time (*p* = 0.000 and *p* = 0.000, respectively)). However, the WMFT time data at T3 showed atypical results as opposed to the other test moments and two outliers were identified (participant 2 and 6) (as presented in Figure 2). Therefore, the median of T3 (median = 136.53) is much higher than the highest median of the other test moments (highest median = 65.82 at T0), creating a significant deterioration between baseline and cessation of training (T3) (*p* = 0.039). When deleting these two participants from the T3 data, the median (adapted median 59.84 (24.34–166.30)) is more in line with the other medians. When looking at the complete time period (baseline T0 to 9 weeks follow-up T4), WMFT time showed a significant improvement (*p* = 0.001). Looking at the different time intervals, the WMFT functional ability showed significant improvement between baseline (T0) and after 6 weeks of training (T3) (*p* = 0.004), baseline and 9 weeks follow-up (*p* = 0.002), and even between cessation of training (T3) and 9 weeks follow-up (T4) (*p* = 0.018).

Patients’ self-perception on execution and satisfaction about their personal goals, measured by the COPM, was low at baseline but improved significantly over time (*p* = 0.000 and *p* = 0.000, respectively).

Regarding perceived performance in activities of daily life, scores on MAM-36 were moderate to good, but improved significantly over time (*p* = 0.000) and within the training period (T0–T3, *p* = 0.007).

For fatigue and trunk impairment, no significant differences were found, but scores on MFIS and TIS increase between baseline and the cessation of training, so participants experienced more fatigue and trunk impairment during the training period but decreased again towards 9 weeks follow-up.

For the AROM in the shoulder joint, a significant improvement was found over time (*p* = 0.000 for shoulder abduction and *p* = 0.001 for shoulder anteflexion).

No adverse events related to the i-ACT training were reported by any of the participants.

### 3.2. Qualitative results

Fourteen patients and eight occupational therapists were interviewed after follow-up assessment. The interview was executed to collect statements regarding the use of i-ACT and gain more in-depth information regarding usability, motivation, credibility and expectancy towards i-ACT.

Both patients and therapists expressed their belief in the i-ACT system as an additional tool in neurological rehabilitation.


*“I learned other stuff than in my regular therapy… I really believe that this in combination with my normal therapies, helped me regain arm functionality”*
*(Patient with stroke)*


*“Technology is the future, but we are still needed. Patients still want a human standing beside them who tells them what is good and what is wrong, or give them tips and tricks”*
*(occupational therapist)*

Patients liked exercising with the system as it was something different than usual.


*“If I’m honest, I secretly liked it (i.e., i-ACT training)”*
*(Person with MS)*


*“Patients tell a lot about their training with i-ACT. They all like doing it, just because it’s something different, something new”*
*(occupational therapist)*

Furthermore, patients felt motivated to perform at their best ability.


*“You have to do your exercises really well, otherwise you get a red dot. If you do your therapy exercises at home, you never know whether you do them correctly or not”*
*(person with stroke)*

Patients as well as therapists also expressed some barriers of the i-ACT. They understood that a prototype was used, but the fact that the avatar was not always moving smoothly was an overall point of discussion.


*“It looks like the little guy is shivering most of the time. It distracts me sometimes”*
*(person with spinal cord injury)*

Another barrier is specifically expressed by the therapists. Five therapists wanted to use the i-ACT more than they did, but due to a lack of time, it was not possible. Additionally, they received a workshop in the beginning, but some refreshment training was necessary when they did not use the i-ACT for a few weeks.


*“After the first workshop, I thought it would be easy. But after that I did not have enough time to get myself acquainted with i-ACT, let alone practice with a patient, and then you forget about certain steps. (…) We need a very intuitive system or more time for training with the system with a professional”*
*(occupational therapist)*

Although therapists were initially critical towards the practicality of the system, after becoming more acquainted with the system, they considered i-ACT to be a user-friendly system with opportunities for additional training in rehabilitation.


*“At least one therapist who works here, should become a specialist in using the i-ACT, then he or she can motivate the colleagues, provide assistance when needed and give workshops every now and then.”*
*(occupational therapist)*

## 4. Discussion

The purpose of i-ACT is to provide additional client-centred task-oriented therapy with feedback towards results and quality of movement. When used additionally to conventional care, patients might increase their exercise and/or task repetitions, and as such, yield positive training effects [52]. Before engaging a large clinical trial, it is necessary to investigate the usability of i-ACT in neurorehabilitation and gather information on patients’ motivation, credibility, and treatment expectancy towards i-ACT (i.e., primary aim of this study).

Earlier studies stated that patients are more motivated when offered virtual rehabilitation exercises [21,23,24,25,26,27,52,53] and consequently might increase therapy adherence [1]. Furthermore, adherence increases when using a client-centred approach [10]. Although i-ACT is not using virtual reality but rather non-immersive VR or augmented reality, Microsoft Kinect and other similar technologies such as Nintendo Wii are commonly categorised as virtual reality. The results from this study showed high values in all subscales of the IMI and a high adherence percentage (i.e., 89.22%), with the exception of the subscale “felt pressure/tension” (medians ≤ 2.2/7). Contrary to the other subscales, these low scores are preferable as people do not want to feel a lot of pressure or tension while exercising. Overall, the IMI results imply that people are motivated to perform exercises with i-ACT, which was also confirmed by the statements in the interviews. These results are in accordance with the study by Lloréns et al. (2015) [53], who found a positive trend on the IMI after intervention with a Kinect-based system for balance recovery after stroke also with lower scores on the subscale “felt pressure/tension” [53]. The high scores in IMI during the i-ACT training period and after 9 weeks follow-up (T4) and the positive statement during the interviews also indicate that patients are able to stay motivated and so training with i-ACT for at least 6 weeks is feasible in neurological rehabilitation.

In contrast, the scores on the MFIS are increasing, indicating more impact of fatigue on physical, cognitive and psychosocial functioning, from T0 to T3 (i.e., 6 weeks training period). At 9 weeks follow-up, the scores are decreased in relation to both T0 (baseline) and T3 (cessation of training). These results can indicate that during training, the additional training with i-ACT had an impact on fatigue but that participants can recover. During the interviews, patients did not mention that they felt more fatigued. Although people experienced more fatigue during the training period, this was neither an influence on patients’ motivation reported by IMI, nor in patients’ performance as reported by WMFT, COPM or MAM. Not only were the patients motivated to use i-ACT, but they also felt that i-ACT is a usable system for additional therapy in neurorehabilitation. These findings were also confirmed in the interview results and are in conformity with the study of Lloréns et al. (2015) and suggest that the system is easy to use and easy to learn [53]. Although the usability results remained good (above the threshold), the scores on the SUS slightly decreased over time. This might be due to the fact that in the beginning of i-ACT training, participants expect some difficulties in working with an unknown technology-based system. Once they have more experience with i-ACT, they may find problems that occur more problematic, such as jittering. Jittering occurs as i-ACT uses the Microsoft Kinect for observing human body movement and streaming this motion data to the application. However, as with many computer vision solutions, it does not always have a perfectly clear signal. If viewed without any form of filtering, this would cause the avatar to jitter of jump around constantly. The strength of the filter determines how much of the original movement is retained and how much passes through to the application’s human body view. The data from both the quantitative assessments and the interviews, suggests that patients believe in the added value of i-ACT for rehabilitation purposes, but they have moderate expectations of a positive treatment outcome when using i-ACT in neurorehabilitation. The latter might be due to the client-centred task-oriented approach. Based on the COPM, patients’ individual goals were trained with i-ACT, and participants reported improvement on these specific tasks, but would probably also expect the same improvements in other tasks.

From previous research, conversations with therapists and board members of rehabilitation centres [21,28,30], and the present research, the demand for affordable, easy-to-use, and easy-to-set-up technology-based systems is evident. Although it is apparent from the interviews that patients and therapists can see the advantages and opportunities of the i-ACT as an additional tool for client-centred training, the options towards the implementation of the system in rehabilitation and/or patients’ homes have yet to be investigated.

The second and third aim of this study was to explore possible effects on functional ability, perceived performance, fatigue, trunk impairment and active range of motion. Regarding functional performance, it can be stated that the overall scores are good, as shown by results on the WMFT. Patients’ upper-limb functional ability progresses well over time as well as range of motion in the shoulder. The upper limb functional ability progress, which was established with the WMFT, is in accordance with the results of previous research with other technology-based interventions [54,55]. There are two notable results. First, there is the very large difference in performance time on WMFT between T3 and the other test moments. This is probably due to the results of five participants who needed more than 2 min to execute tasks or could not complete one or more tasks at T3. The maximum score of 120 (seconds) is then given [41]. Additionally, two outliers were observed, i.e., participant 2 and 6. These two patients reported fatigue at test moment T3 due to more difficult and heavier conventional care. However, taking out the results of T3 and looking at the difference between T0 versus T4, a significant difference of *p* = 0.001 was found in favour of T4. These results are in contrast to other research, where an effect was found on the WMFT-FAS part, but no remarkable effect was found on the time component of the WMFT [54,55].

In general, the good-to-high scores on the primary outcome measures (i.e., IMI, SUS and CEQ) and WMFT functional ability scale, MAM-36 and COPM, are in favour of using i-ACT as an additional client-centred task-oriented therapy tool. Furthermore, as no adverse events were reported towards i-ACT and considering the positive statements of both patients and therapists in the interviews, additional training with i-ACT might increase patient’s motivation and adherence towards therapy and even increase functional performance in ADL.

In total, only 17 neurological patients participated in the study. During the training period, four persons were discharged. These persons were contacted for T3 and T4 assessments. At T3, these people did not wish to participate, but one of these persons agreed to participate at follow-up. The main reason to not participate in the assessments was the recollection towards the rehabilitation, i.e., people wished to be left alone once discharged from the rehabilitation programme as they did not want to be reminded of that time. Other studies with rehabilitation technology in neurorehabilitation have similar numbers of participants [1,15,21]. Furthermore, the sample consisted of a very diverse population, but mainly male persons and persons with stroke. Looking at individual data of the participants with spinal cord injury, craniocerebral trauma, and Parsonage Turner syndrome, the results over time are in line with the results of participants with stroke. Although the numbers of these patient groups are very limited, we can assume that the i-ACT training is also suitable, useful and motivational for these patient groups. However, more research with these patient groups is needed. Furthermore, the time since stroke was about 5 months and could have influenced the results. It is known that in the first 3 months after stroke, the most motor recovery occurs [56]. When looking into the separate data of stroke persons, six persons were considered to be in the late subacute stage (3–6 months) and five persons were in the chronic stage (>6 months) [56]. Due to the low numbers, no statistical analysis was performed, but the raw data were compared. However, no significant or remarkable differences were found. No statements can be made regarding persons in the acute stage of stroke, but this could be addressed in future research. Not only was the numbers of participants limited, but there was also no control group. This was a practical choice as this is an exploratory study to look for possible indications to perform a randomized controlled trial in the future. Consequently, no conclusions can be made regarding the efficacy of the i-ACT system. Although its efficacy was not the focus of this cohort study, it is recommended that in future research with i-ACT, a control group is added to the study design.

The qualitative data analysis was secondary to the quantitative analyses and used as data-triangulation to gain more in-depth insight in the users’ experiences (i.e., usability, motivation, credibility and expectancy towards training with i-ACT). The reported findings in this exploratory cohort study have to be considered as an indication for future research. This study found good to very good, and even significantly positive results on motivation (IMI), usability (SUS), credibility and expectancy of therapy (CEQ) with i-ACT, and functional performance (WMFT). The motivation to use i-ACT is high and there is even an indication of a better functional ability as seen in the WMFT, COPM and MAM-36. These outcome measures are classified under ICF activity level and thus data from these secondary outcome measures mainly show (significant) improvement on patient’s activity level. However, as no control group was included, no comparison can be made between conventional care, conventional care with additional i-ACT training, or a control group with the same amount of training as an experimental group with conventional and i-ACT training. Therefore, the findings in this study have to be handled with care and seen in the light of an exploratory study as a motivation to perform a higher-level clinical study (i.e., randomized controlled trial).

## 5. Conclusions

The i-ACT system is a unique system as it is sensor-based, low-cost, and integrates a client-centred and task-oriented approach. The i-ACT is considered to be usable by patients and therapists. It is considered to be an added value within client-centred activity-based training in neurological rehabilitation as patients’ motivation, credibility, and adherence to i-ACT therapy are high. Patients believe they can benefit from additional therapy with i-ACT for functional training and the data show encouraging results towards the increase of functional upper limb performance. As i-ACT is a promising system for physical rehabilitation in neurological diseases, further research into the clinical efficacy of training with the system is warranted.

## Figures and Tables

**Figure 1 ijerph-18-07641-f001:**
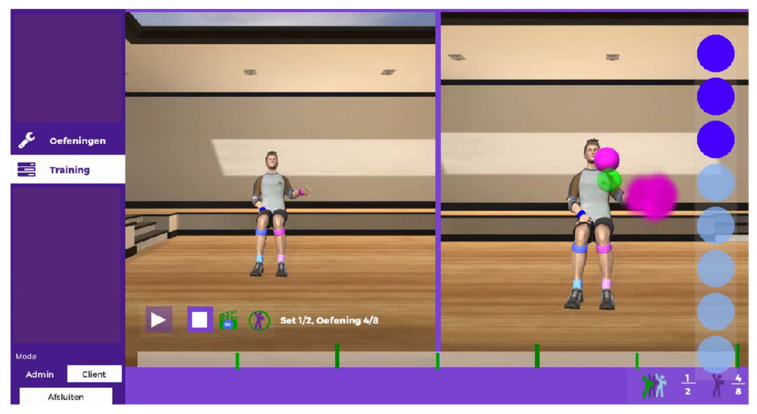
Patient interface during performance of the exercise “drinking from a cup”. The avatar on the left represents the therapist (i.e., recorded movement) while the avatar on the right represents the patient. The green dot is the stability point for restriction of compensational movement. The pink dots are the targets for the right hand.

**Figure 2 ijerph-18-07641-f002:**
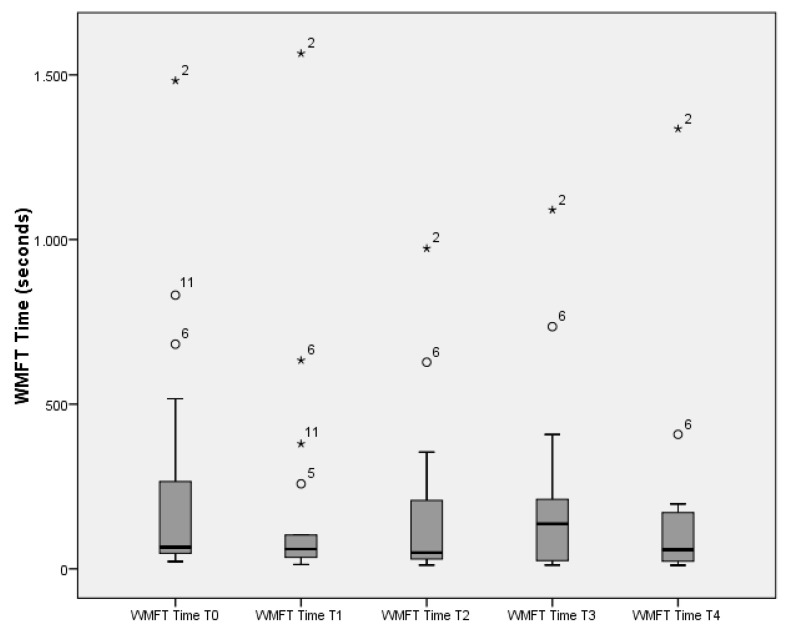
Plotted data of WMFT time.

**Table 1 ijerph-18-07641-t001:** Overview of assessments and assessment times.

	T0			T1	T2	T3	T4
	T0a	T0b	T0c				
	−2 Weeks	−1 Week	Week 0	Week 2	Week 4	Week 6	Week 15
Primary outcome measures			COPM			COPM	COPM
			IMI ^1^	IMI	IMI	IMI	IMI
			SUS ^1^	SUS	SUS	SUS	SUS
			CEQ ^1^	CEQ	CEQ	CEQ	CEQ
	WMFT	WMFT	WMFT	WMFT	WMFT	WMFT	WMFT
Secondary outcome measures	MAM-36	MAM-36	MAM-36	MAM-36	MAM-36	MAM-36	MAM-36
	TIS	TIS	TIS	TIS	TIS	TIS	TIS
	MFIS	MFIS	MFIS	MFIS	MFIS	MFIS	MFIS
	AROM	AROM	AROM	AROM	AROM	AROM	AROM

COPM: Canadian occupational performance measure; IMI: intrinsic motivation inventory; SUS: system usability scale; CEQ: credibility/expectancy questionnaire; WMFT: Wolf motor function test; MAM-36: manual ability measure; TIS: trunk impairment scale; MFIS: modified fatigue impact scale; AROM: active range of motion; ^1^: assessed after one training session with i-ACT.

**Table 2 ijerph-18-07641-t002:** Patient and therapist characteristics.

	Patients	Therapists
N	17	8
Age (years) ^1^ Gender (male/female) Diagnosis (time (months) since diagnosis) ^1^ Stroke Multiple sclerosis Spinal cord injury Craniocerebral trauma Parsonage Turner syndrome	57.2 ± 16.3 13/4 11 (5.00 ± 2.14) 2 (289.00 ± 151.32) 1 (5.00) 2 (16.00 ± 9.90) 1 (3.00)	35.6 ± 4.7 0/8

^1^ Data represented as mean ± standard deviation.

**Table 3 ijerph-18-07641-t003:** Results of all outcome measures.

	Baseline	Training Period			Follow-Up	Within Group						
	T_0_	T_1_	T_2_	T_3_	T_4_	Over Time *	T_0_–T_3_ ^£^	Effect Size	T_0_–T_4_^£^	Effect Size	T_3_–T_4_ ^£^	Effect Size
	*n* = 17	*n* = 17	*n* = 17	*n* = 13	*n* = 14	T_0_–T_4_		T_0_–T_3_		T_0_–T_4_		T_3_–T_4_
IMI (range 1–7)												
Interest/enjoyment	6.00 (5.07–6.64)	5.86 (4.43–6.50)	6.00 (5.15–6.50)	6.00 (5.22–6.29)	5.93 (5.54–6.50)	*p* = 0.413	*p* = 0.875		*p* = 0.506		*p* = 0.400	
Perceived competence	5.17 (3.75–5.83)	5.33 (4.33–5.67)	5.67 (4.84–6.00)	5.83 (4.59–6.17)	5.67 (5.13–6.04)	*p* = 0.025	*p* = 0.017	r = 0.434	*p* = 0.035	r = 0.378	*p* = 0.789	
Effort	6.00 (5.50–6.70)	6.00 (5.40–6.40)	6.00 (5.40–6.70)	6.20 (5.20–6.70)	6.4 (5.70–7.00)	*p* = 0.543	*p* = 0.345		*p* = 0.806		*p* = 0.232	
Felt pressure/tension	2.00 (1.10–3.50)	2.20 (1.50–2.80)	1.80 (1.20–2.80)	2.00 (1.10–2.30)	2.20 (1.35–2.85)	*p* = 0.682	*p* = 0.142		*p* = 0.325		*p* = 0.422	
Value/usefulness	6.00 (5.15–6.64)	6.00 (5.22–6.57)	6.29 (5.72–6.71)	6.00 (4.86–6.79)	6.36 (5.36–6.78)	*p* = 0.086	*p* = 0.484		*p* = 0.099		*p* = 0.247	
Relatedness	5.80 (5.10–6.20)	5.80 (4.90–6.80)	6.00 (5.80–6.60)	6.00 (5.30–6.60)	5.70 (5.20–7.00)	*p* = 0.884	*p* = 0.972		*p* = 0.504		*p* = 0.944	
SUS (range 0–100)	77.50 (71.25–85.00)	77.50 (61.25–81.25)	75.00 (58.75–86.25)	75.00 (63.75–90.00)	73.75 (64.38–92.50)	*p* = 0.174	*p* = 0.562		*p* = 0.975		*p* = 0.858	
CEQ (range 0–27)												
Credibility	22.00 (18.00–25.00)	23.00 (17.00–25.00)	23.00 (18.00–25.50)	23.00 (16.00–24.50)	24.00 (20.25–25.00)	*p* = 0.176	*p* = 0.655		*p* = 0.302		*p* = 0.020	r = 0.447
Expectancy	16.90 (10.80–20.10)	16.00 (12.80–22.45)	18.80 (13.15–21.85)	17.70 (11.25–21.00)	15.85 (10.30–19.83)	*p* = 0.563	*p* = 0.959		*p* = 0.529		*p* = 0.308	
WMFT												
Functional ability (range 0–75)	63.67 (37.25–71.00)	63.00 (46.00–72.50)	68.00 (44.00–74.00)	70.00 (43.00–74.00)	67.00 (53.50–74.25)	*p* = 0.000	*p* = 0.004	r = 0.523	*p* = 0.002	r = 0.547	*p* = 0.018	r = 0.455
Performance time (sec)	65.82 (40.72–390.78)	59.91 (31.95–180.27)	49.01 (28.76–269.16)	136.53 (24.34–309.56)	57.96 (23.23–177.30)	*p* = 0.000	*p* = 0.039	r = −0.376	*p* = 0.001	r = −0.581	*p* = 0.046	r = −0.383
COPM (range 0–10)												
Performance	3.00 (0.85–3.75)			7.00 (5.15–7.80)	7.00 (4.68–8.00)	*p* = 0.000	*p* = 0.002	r = 0.559	*p* = 0.001	r = 0.581	*p* = 0.969	
Satisfaction	3.00 (0.15–4.30)			6.00 (4.00–7.80)	7.10 (5.20–8.50)	*p* = 0.000	*p* = 0.003	r = 0.536	*p* = 0.001	r = 0.592	*p* = 0.071	
MAM-36 (range 0–144)	73.00 (35.15–102.85)	79.00 (34.00–109.00)	87.00 (44.00–118.50)	79.00 (39.50–122.0)	97.00 (38.50–127.00)	*p* = 0.000	*p* = 0.007	r = 0.493	*p* = 0.001	r = 0.592	*p* = 0.052	
MFIS (range 0–84)	19.00 (4.50–41.50)	30.00 (6.00–53.50)	24.00 (3.00–42.00)	24.00 (5.50–43.50)	15.50 (2.75–41.75)	*p* = 0.254	*p* = 0.824		*p* = 0.432		*p* = 0.160	
TIS (range 0–23)	19.00 (13.85–21.85)	21.00 (12.50–22.00)	21.00 (16.00–23.00)	21.00 (11.00–23.00)	20.50 (6.00–23.00)	*p* = 0.342	*p* = 0.514		*p* = 0.514		*p* = 0.833	
AROM (degrees)												
Shoulder abduction	88.30 (67.50–162.50)	90.00 (70.00–150.00)	100.00 (75.00–155.00)	100.00 (70.00–170.00)	102.50 (68.75–160.00)	*p* = 0.000	*p* = 0.109		*p* = 0.100		*p* = 0.167	
Shoulder anteflexion	95.00 (80.00–147.50)	100.00 (80.00–137.50)	115.00 (82.50–147.50)	100.00 (82.50–157.50)	112.50 (80.50–162.50)	*p* = 0.001	*p* = 0.071		*p* = 0.011	r = 0.459	*p* = 0.019	r = 0.453

Values are median (interquartile range). * *p* < 0.05 in Friedman′s ANOVA; £ *p* < 0.05 in Wilcoxon signed-rank test. T0: baseline; T1: after 2 weeks of training; T2: after 4 weeks of training; T3: after 6 weeks of training; T4: 8–10 weeks follow-up; IMI: intrinsic motivation inventory; SUS: system usability scale; CEQ: credibility/expectancy questionnaire; WMFT: Wolf motor function test; COPM: Canadian occupational performance measure; MAM-36: manual ability measure; MFIS: modified fatigue impact scale; TIS: trunk impairment scale; AROM: active range of motion. The WMFT, MAM-36, TIS and AROM were tested on the most affected side.

## Data Availability

The data in this study are available on reasonable request from the corresponding author (E.K.). The data are not publicly available as they contain information that could compromise the privacy of research participants.

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
