# Peer review of "Motivation, Usability, and Credibility of an Intelligent Activity-Based Client-Centred Training System to Improve Functional Performance in Neurological Rehabilitation: An Exploratory Cohort Study"

_ijerph, 2021, doi:10.3390/ijerph18147641_

Round 1

Reviewer 1 Report

This study is very interesting due to the low number of studies with  technology-based training system applied to neuroreabilitation and combined with a client-centred task-oriented approach. Also the game-based virtual reality intervention is a promising modality for motor rehabilitation. 

Some limitations are mentioned in the text, such as the small sample size, tand the non-existence of a control group. However, others like the effect size is not presented. All of these aspects reduce the precision level of the findings and increase the possibility of selection bias.

Author Response

Point 1: This study is very interesting due to the low number of studies with  technology-based training system applied to neuroreabilitation and combined with a client-centred task-oriented approach. Also the game-based virtual reality intervention is a promising modality for motor rehabilitation. 

Some limitations are mentioned in the text, such as the small sample size, tand the non-existence of a control group. However, others like the effect size is not presented. All of these aspects reduce the precision level of the findings and increase the possibility of selection bias.

Response 1: The authors thank the reviewer for his/her feedback. The effect size is indeed not represented as it is a pilot cohort study to look into the possible effects and not towards the size of the effect. Nevertheless, the effect sizes are added in table 3 for the significant within group differences (Please see the attachment "Table 3 - Total scores of all outcome measures"). 

Reviewer 2 Report

The paper entitled “Motivation, usability, and credibility of an intelligent activity-based client-centred training system to improve functional performance in neurological rehabilitation: an exploratory cohort study” describes a mixed-method study aimed to evaluate several aspects of a low-cost client centred-task-oriented intervention.

The study is of great interest to IJERPH readers. It provides timely information and the methods are rigorous. The findings are well discussed and conclusions are supported by the results. I only have some minor comments for the authors:

Participants (lines 71-86). It would be recommended to include a sample size calculation in order to provide an estimation of the confidence interval and statistical power of the final sample.

Outcome measures (lines 146-190). The authors have done a remarkable job assessing so many variables, but it its difficult to read. I highly recommend authors to describe each measure in different paragraphs (e.g., 2.4.1. The IMI; 2.4.2. The SUS, etc.).

Table 3. The title is not shown and the table is shown incomplete.

Author Response

The paper entitled “Motivation, usability, and credibility of an intelligent activity-based client-centred training system to improve functional performance in neurological rehabilitation: an exploratory cohort study” describes a mixed-method study aimed to evaluate several aspects of a low-cost client centred-task-oriented intervention.

The study is of great interest to IJERPH readers. It provides timely information and the methods are rigorous. The findings are well discussed and conclusions are supported by the results. I only have some minor comments for the authors:

Point 1: Participants (lines 71-86). It would be recommended to include a sample size calculation in order to provide an estimation of the confidence interval and statistical power of the final sample.

Response 1: The authors thank the reviewer for his/her critical feedback. As this is a first longitudinal trial, the chosen sampling strategy was based on the accessibility and proximity to the research and participants were selected in an ad hoc fashion. This is known as convenience sampling and is the norm within developmental science [1]. Therefore no sample size calculations were performed for this study, but the results from this study were used to set up a randomised controlled trial.

Reference:
1.         Jager J, Putnick DL, Bornstein MH. II. More than just convenient: The scientific merits of homogeneous convenience samples. Monographs of the Society for Research in Child Development. 2017;82(2):13-30.

Point 2: Outcome measures (lines 146-190). The authors have done a remarkable job assessing so many variables, but it its difficult to read. I highly recommend authors to describe each measure in different paragraphs (e.g., 2.4.1. The IMI; 2.4.2. The SUS, etc.).

Response 2: We thank the reviewer for this feedback and we decided to describe each measurement in different paragraphs with the name of the measurement in bold to highlight the subject of the paragraph.

Point 3: Table 3. The title is not shown and the table is shown incomplete.

Response 3: We submitted table 3 as a separate document due to the fact that the table needs to be presented in landscape view. We saw indeed that table 3 was now inserted in the text without adapted lay-out. For this we apologise. We inserted table 3 now as a figure-like object and also submitted table 3 as an extra attachment to this response (please see the attachment “ Table 3 – Total scores of all outcome measures”.

Reviewer 3 Report

The study aims to evaluate the motivation, usability, and credibility of using a previously developed, Microsoft Kinect-based rehabilitation system. The authors explain the methods of the study transparently. While the study design is strong, the manuscript in its current form could be stronger if some areas of concern are addressed. Specific comments are listed below.

  1. The study is based on using a previously developed rehabilitation system. While the development of the i-ACT system is described in another manuscript that is cited, the current manuscript will benefit if an overview of the system is added to the Introduction section.
  2. The Introduction also lacks a comprehensive comparison of other rehabilitation systems being used from the literature and hence the ‘research gap’ is not established well.
  3. Re-format table 3 so that all columns are clearly visible.  
  4. Provide more information on the type of exercises performed as part of the training.
  5. Please note in the limitations that detailed qualitative analysis was not performed.
  6. The diagnosis of the patients is quite different with some participants in the acute phase and some in the chronic phase. Discuss this limitation further and include how this may have affected the results. For example, how many stroke patients were in the acute phase? Did participants in the acute phase have higher scores across outcome measures compared to chronic stroke patients? What about differences in outcome measures based on the diagnosis. Even though the sample size is not big enough to perform statistical analysis, discussing the differences will be valuable.

Author Response

The study aims to evaluate the motivation, usability, and credibility of using a previously developed, Microsoft Kinect-based rehabilitation system. The authors explain the methods of the study transparently. While the study design is strong, the manuscript in its current form could be stronger if some areas of concern are addressed. Specific comments are listed below.

Point 1: The study is based on using a previously developed rehabilitation system. While the development of the i-ACT system is described in another manuscript that is cited, the current manuscript will benefit if an overview of the system is added to the Introduction section.

Response 1: The authors would like to thank the reviewer for his/her critical feedback as they are convinced it will improve the manuscript. As for the description of the used i-ACT system, we added the following sentences in the introduction:

With i-ACT, therapists can record a movement based on the individually set goals of a persons with CNS central nervous system disorder. Next, it is possible to implement certain training parameters such as amount of repetitions, amount of rest, number and location of the targets. During training, the patient has a mirrored view of himself represented as an avatar and is stimulated to perform the tasks or activities presented on the screen. A more detailed description is provided in the method section.“  

For a more elaborated description of i-ACT we would like to refer the reviewer to the method section under “2.3 Apparatus”. In this section we explain the main features of the apparatus and provide a figure about the patient’s interface. Going into more detail, would mean more technical information and more information about how the system works, which is elaborately explained in the referred article regarding the development of i-ACT, and would decrease the objective of the present study.

Point 2: The Introduction also lacks a comprehensive comparison of other rehabilitation systems being used from the literature and hence the ‘research gap’ is not established well.

Response 2: The authors added some information in the introduction and refer to the track changes in the revised document. In the introduction we state that a client-centred task-oriented approach is important but not easily implemented in neurorehabilitation due to costs and time restrictions. A possible solution might be technology, but robotics and low-cost technologies have their restrictions regarding integrating a client-centred task-oriented approach. The authors did not want to provide a summary of varies technologies and thus referred to several systematic reviews that looked into the use of robotics or low-cost technologies in neurorehabilitation. None of the systematic reviews mention the use of a technology with a client-centred and task-oriented approach, although important in neurological rehabilitation. And thus we developed a low-cost technology-based system that integrates a client-centred task-oriented approach (i.e. i-ACT). After the development, it is necessary to test the system in the clinical field. And that is the content of this present study: the first longitudinal study with i-ACT in neurorehabilitation.  

Point 3: Re-format table 3 so that all columns are clearly visible.  

Response 3: We submitted table 3 as a separate document due to the fact that the table needs to be presented in landscape view. We saw indeed that table 3 was now inserted in the text without adapted lay-out. For this we apologise. We inserted table 3 now as a figure-like object and also submitted table 3 as an extra attachment to this response (please see the attachment “ Table 3 – Total scores of all outcome measures”.

Point 4: Provide more information on the type of exercises performed as part of the training.

Response 4: An overview of exercises was too elaborate to provide as individualised goal setting was performed for each participant (i.e. client-centred approach). A few examples of exercises are provided in “2.2 Study protocol”.  

Point 5: Please note in the limitations that detailed qualitative analysis was not performed.

Response 5: The authors agree with the reviewer and added the suggested limitation in the discussion in a separate paragraph: “The qualitative data analysis was secondary to the quantitative analyses and used as data-triangulation to gain more in depth insight in the users’ experiences (i.e. usability, motivation, credibility and expectancy towards training with i-ACT).”

Point 6: The diagnosis of the patients is quite different with some participants in the acute phase and some in the chronic phase. Discuss this limitation further and include how this may have affected the results. For example, how many stroke patients were in the acute phase? Did participants in the acute phase have higher scores across outcome measures compared to chronic stroke patients? What about differences in outcome measures based on the diagnosis. Even though the sample size is not big enough to perform statistical analysis, discussing the differences will be valuable.

Response 6: The authors did not implement a descriptive analysis of acute versus chronic stroke as there were no persons involved who were in the acute stage. However, we considered the feedback of the reviewer and added a paragraph in the discussion in which we briefly describe the data of persons in the late subacute stage of stroke and persons in the chronic stage of stroke: “When looking into the separate data of stroke persons, 6 persons were considered to be in the late subacute stage (3-6 months) and 5 persons were in the chronic stage (> 6 months). Due to the low numbers, no statistical analysis was performed, but the raw data were compared. However, no significant or remarkable differences were found. No statements can be made regarding persons in the acute stage of stroke, but this could be addressed in future research.”

Reviewer 4 Report

I believe that the topic of the study should be of interest to the readers in International Journal of Environmental Research and Public Health. The present manuscript is an applied article with regard to neurological rehabilitation. The authors must interpret their findings in the context of supporting/not supporting existing literature and provide neurological rehabilitation applicability. However, I believe the authors need to revise some of the contents of the paper.

  1. The research uses intelligent activity-based customer-centric task-oriented training (i-ACT), There has not been any research in the past to ensure reliability. There are doubts about the choice of participants because neurological rehabilitation patients existed highly variable.

  1. The research method is too complicated, and the use of many questionnaires may affect the authenticity of the experiment participants?

  1. The two groups of experimental participants are uneven in number, and the doubts about the number of participants are too small, and the gender difference is too large. Is it possible that the results of the study may be affected?

  1. Some participants have different clinical symptoms, such as multiple sclerosis, spinal cord injury, craniocerebral trauma, Parsonage turner syndrome, will it affect the results of the study?

  1. The content of Table 3 is incomplete.

  1. Why not find healthy as the control group for comparison?

  1. The conclusion mentioned "The i-ACT system is regarded to be usable by patients and therapists". However, other systems also have the same functions. How to highlight the speciality and contribution of i-ACT?

Author Response

I believe that the topic of the study should be of interest to the readers in International Journal of Environmental Research and Public Health. The present manuscript is an applied article with regard to neurological rehabilitation. The authors must interpret their findings in the context of supporting/not supporting existing literature and provide neurological rehabilitation applicability. However, I believe the authors need to revise some of the contents of the paper.

Point 1: The research uses intelligent activity-based customer-centric task-oriented training (i-ACT), There has not been any research in the past to ensure reliability. There are doubts about the choice of participants because neurological rehabilitation patients existed highly variable.

  1. Bonnechere B, Jansen B, Salvia P, Bouzahouene H, Omelina L, Moiseev F, et al. Validity and reliability of the Kinect within functional assessment activities: comparison with standard stereophotogrammetry. Gait & posture. 2014;39(1):593-8.
  2. Clark RA, Pua YH, Oliveira CC, Bower KJ, Thilarajah S, McGaw R, et al. Reliability and concurrent validity of the Microsoft Xbox One Kinect for assessment of standing balance and postural control. Gait & posture. 2015;42(2):210-3.
  3. Knippenberg E, Van Hout L, Smeets W, Palmaers S, Timmermans A, Spooren A. Developing an intelligent activity-based client-centred training system with a user-centred approach. Technology and health care : official journal of the European Society for Engineering and Medicine. 2020;28(4):355-68.

Point 2: The research method is too complicated, and the use of many questionnaires may affect the authenticity of the experiment participants?

Response 2: The objective of the present study is to explore different aspects of a newly developed rehabilitation technology. These aspects are 1) user’s experience (i.e. usability of the new system, credibility and treatment expectancy of this new system, and the users’ motivation when using this new system), and 2) whether the system has any objectifiable positive effect for the user. In order to be able to get a comprehensive overview, a mixed method was chosen. The first aspect was primarily done by using standard questionnaires (i.e. SUS, CEQ and IMI) and additional information on this user experience was gathered by interviewing the users. Hence a mixed method design.       
To estimate the functional improvements (i.e. the second aspect), we aimed to evaluate on the different levels of the International Classification of Functioning, Disability and Health (ICF), i.e. on ICF Function level (i.e. AROM), on ICF Activity level – Capacity (WMFT  for upper extremity activity ad TIS for trunk activity), and ICF Activity level -performance (MAM-36 for perceived performance on 36 standard activities and COPM for perceived performance on individualised goals).

Point 3: The two groups of experimental participants are uneven in number, and the doubts about the number of participants are too small, and the gender difference is too large. Is it possible that the results of the study may be affected?

Reference:
1.         Jager J, Putnick DL, Bornstein MH. II. More than just convenient: The scientific merits of homogeneous convenience samples. Monographs of the Society for Research in Child Development. 2017;82(2):13-30.

Point 4: Some participants have different clinical symptoms, such as multiple sclerosis, spinal cord injury, craniocerebral trauma, Parsonage turner syndrome, will it affect the results of the study?

Point 5: The content of Table 3 is incomplete.

Response 5: We submitted table 3 as a separate document due to the fact that the table needs to be presented in landscape view. We saw indeed that table 3 was now inserted in the text without adapted lay-out. For this we apologise. We inserted table 3 now as a figure-like object and also submitted table 3 as an extra attachment to this response (please see the attachment “ Table 3 – Total scores of all outcome measures”.

Point 6: Why not find healthy as the control group for comparison?

Response 6: Because healthy persons would perform a perfect score in each of the measurements. And the main objective of this present study was to look into the usability, credibility and treatment expectancy of persons with neurological disorders towards additional training with the i-ACT system. And not to look into a possible difference between persons with a neurological disorder and healthy persons. The next step is to perform a randomised controlled trial with persons with neurological disorders in which the intervention group receives treatment as usual and i-ACT training and the control group solely receives treatment as usual.

Point 7: The conclusion mentioned "The i-ACT system is regarded to be usable by patients and therapists". However, other systems also have the same functions. How to highlight the speciality and contribution of i-ACT?

Response 7: To the authors’ knowledge, there is no other affordable technology-based system that incorporates a client-centred and task-oriented approach. The fact that i-ACT comprises of a low-cost off-the-shelf available motion detection system (i.e. Microsoft Kinect camera) in combination with a self-developed software programme in which client-centredness is fully involved, is unique to the authors’ knowledge. This feature, together with other settings and features, is introduced in the introduction section and further described in the method section. But we agree that this is not highlighted in the conclusion, so we added the following sentence in the conclusion: “The i-ACT system is a unique system as it is sensor-based, low cost, and integrates a client-centred and task-oriented approach.”

Round 2

Reviewer 3 Report

The authors have addressed all comments satisfactorily. The findings presented in the study, significantly add to the existing scientific literature in this domain. 

Author Response

The authors thank the reviewer for his/her feedback and appreciate the comments.

Reviewer 4 Report

The author did not respond to point 4. Ask author to respond on this point.

Author Response

The authors apologise to the reviewer for the inconvenience.

Our response to point 4 was as follows:
We looked into the results of the participants with these different clinical diagnoses and compared them with the results of the persons with stroke. Based on observation of individual results, we could not see any tendencies that the different patient groups performed different as opposed to the persons with stroke. This was also mentioned in the discussion. Due to the low number of these participants with other clinical diagnoses than stroke, no statistical analysis was performed, and further advanced research is necessary to determine whether all these groups respond similar or not, to the additional training with i-ACT.
The following text was added in the discussion section with regards to this comment:
"When looking into the separate data of stroke persons, 6 persons were considered to be in the late subacute stage (3-6 months) and 5 persons were in the chronic stage (>6 months) [56]. Due to the low numbers, no statistical analysis was performed, but the raw data were compared. However, no significant or remarkable differences were found. No statements can be made regarding persons in the acute stage of stroke, but this could be addressed in future research. "

Round 3

Reviewer 4 Report

This article after modification by the author. I believe that the topic of the study should be of interest to the readers in the International Journal of Environmental Research and Public Health.